# Impact of distance and vertical placement of distal implants on bone mechanics in bar-retained overdenture treatment: A 3D finite element study

**Danesh Ghahramanimarangalou**[1], **Nurullah Türker**[2], **Cennet Neslihan Eroğlu**[1]*

1 Department of Oral and Maxillofacial Surgery, Faculty of Dentistry, Akdeniz University, Antalya, Turkey
2 Department of Prosthodontics, Faculty of Dentistry, Akdeniz University, Antalya, Turkey

❧ These authors contributed equally to this work.
* neslihanakca2003@yahoo.com

## Abstract

### Objective

In bar-retained, four-implant supported overdentures, implant positioning influences load transfer to the bone. Variations in distance and vertical alignment between implants may alter stress and strain distribution, affecting long-term outcomes. This study evaluates how the distance and vertical position of the distal implant relative to its neighboring implant affect stress and strain in the surrounding bone using 3D finite element analysis.

### Materials and methods

Four groups were formed by shifting the neighboring implant mesially at 1 mm intervals, creating inter-implant distances of 3.8, 4.7, 5.6, and 6.5 mm. Each group included seven subgroups, where the distal implant was vertically positioned 1, 2, or 3 mm coronally or apically relative to its neighbor. A vertical load of 100 N was applied to the bar's cantilever. Von Mises stress, maximum and minimum principal stresses, and strain values were calculated for 28 models.

### Results

Maximum compressive stress and strain were observed at the distal aspect of the cortical bone surrounding the distal implant, averaging -49 Mpa (megapascal) and 3100 με (microstrains), respectively. Apical positioning of the distal implant reduced compressive stress and strain by up to 22%, whereas coronal positioning produced similar or slightly lower compressive values but increased tensile stress and strain in the mesial cortical bone around the neighboring implant (maximum 25.7 MPa and

**Data availability statement:** All relevant data are within the manuscript and its Supporting Information files.

**Funding:** The author(s) received no specific funding for this work.

**Competing interests:** The authors have declared that no competing interests exist.

1578 µε). Increasing the interimplant distance consistently elevated both stress and strain in the surrounding bone.

## Conclusion

Within the limitations of finite element analysis, minor vertical deviations in distal implant positioning due to bone irregularities or defects do not substantially increase cortical bone stress or strain. However, greater horizontal spacing between implants may result in higher biomechanical loads, emphasizing the importance of careful implant positioning for long-term success.

## Introduction

The increase in life expectancy, and the subsequent increase in edentulousness has led to a growing preference for dental implant treatment options [1]. In cases of long term edentulism, particularly in patients using conventional removable dentures alveolar bone resorption is commonly observed [2]. This bone loss in alveolar bone volume is especially pronounced in the posterior region making implant treatments in this area quite challenging [3,4]. Bone resorption between the mental foramina in the mandible is slower than in the posterior region, and removable prosthesis treatments supported by at least two implants applied to this region may be preferred in cases with excessive bone resorption. Mandibular implant-supported overdenture treatment provides better stability, retention, function, and patient satisfaction compared to conventional removable dentures [5]. The long-term success of this treatment option has been confirmed by many studies [6–11].

In mandibular overdenture treatment, the prosthesis is usually supported by 2, 3, 4 or 5 implants. Overdentures supported by four or more implants have advantages such as providing better retention and allowing for distal cantilevers in bar-retained treatments. In addition, overdenture treatments over 4 implants may be preferred in the presence of dentition in maxilla, in cases where the bone is thin and short, in the presence of sensitive mucosa, in the presence of increased occlusal forces and in cases with high muscle attachments [5]. In four-implant-supported mandibular overdenture planning, it is recommended that the bone of the anterior to the mental foramen is divided into 5 equal parts [1]. But in cases of unfavorable bone or when a new implant replaces a failed implant, the implant may not be placed in the ideal position [12]. Also, in implant-supported removable prosthesis treatments, it is generally preferred that the implants be at the same occlusal level [1]. But this may not always be possible due to the order in which the teeth are extracted and the resulting bone irregularities and bone defects.

One of the most common causes of late failure of osteointegrated implants is bone loss in the alveolar ridge due to excessive stress that occurs around the implant, thus it is important that we have methods of analyzing the biomechanical factors affecting the implants [13–15]. Finite element analysis (FEA), strain gauge, photoelasticity and digital image correlation are the most widely used methods for biomechanical

analysis of dental implants with a noninvasive technique [16]. This method, which allows for the examination of structures in the desired sections, is easily repeatable and requires less time compared to other biomechanical analysis methods [17]. It has been widely used by researchers in dental implant simulations [18–20].

The primary challenge in simulating the mechanical behavior of dental implants lies in modeling the bone tissue's response to applied forces. To achieve this, certain assumptions must be made. Due to the complexity of bone mechanics and its interaction with the implant system, researchers are often required to introduce simplifications. Some of these assumptions significantly impact the accuracy of finite element analysis [21].

Finite element analysis offers advantages such as the ability to study complex geometries and materials, easily modify the system, and explore multiple alternatives. However, it also has limitations, including the subjectivity of modeling assumptions and the lack of guaranteed accuracy in the results. Therefore, findings obtained through this method should be validated using other techniques [21]. Nevertheless, studies have reported that the outcomes of finite element analyses used in implant biomechanics are consistent with those obtained from in vitro experiments [22]. The aim of this study is to investigate the effects of unideal positioning and vertical misalignment of the most distal implants in 4-implant-supported overdentures level, on the stress and strain in the bone, using 3D FEA.

## Materials and methods

### Modeling and analysis software

Blender 2.82 (Blender Foundation, community) was used for creating three-dimensional geometry, Salome 9.8.0 (Électricité de France, Open Cascade, French Alternative Energies and Atomic Energy Commission) was used for creating three-dimensional solid model, and for the Analysis CalculiX 2.20 (Guido Dhondt,) for analysis. Klaus Wittig) was used.

### Creation of the meshes

According to the geometry of the models, meshes with tetrahedral elements were created using the NETGEN algorithm. Repeated mesh refinement was performed in regions with a high probability of occurrence of singularity. The number of nodes and elements are similar among the working models, and after refinement, each model consists of approximately 156,000 nodes and 32,000 tetrahedral elements on average. The overall geometry of the mandibular model, the implant-bar assembly, and the boundary conditions applied during the simulation are illustrated in Fig 1.

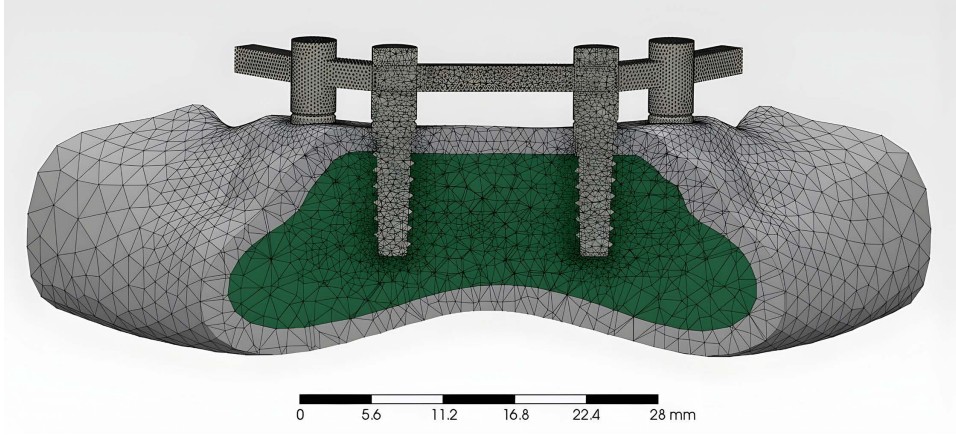

**Fig 1. Finite element model of the mandibular structure.**

The meshes were exported in UNV format from the Salome software and transferred to the CalculiX software. Here, the analysis was carried out by determining the boundary conditions and material properties.

## Creation of the models

The morphology of the bone was modeled in Blender, using a study on the morphology of the mandibular bone, including the part of the toothless lower jaw anterior to the mandibular third molar [23]. To reduce specific variables that would affect the result, the bone was modeled symmetrically with respect to the sagittal plane and without bone irregularities in the parts where the implants would be placed. Then, according to each study group, the height of the bone in the area where the most distal implant on the left side would be placed was modified to have the desired dimensions.

Cortical bone with an average thickness of 2 mm was modeled separately for all groups [23,24].

The coordinates of the cross sections of these models were transferred to the Salome program, and the volumes were obtained.

The implant geometry was modeled according to the most common type of implants used today which are the screw type. The implants were modeled in the Salome software with a length of 10 mm and a diameter of 3.5 mm. In order to simplify the model and because the stresses of prosthetic parts are not within the scope of this study, abutment parts were modeled combined with bars and implants (Fig 2). The lengths of the abutments were chosen to ensure that the bar was placed at least 3 mm away from the bone and parallel to the occlusal plane [25]. The bar is modeled with a size of 2 mm x 2 mm and a distal cantilever of 8 mm [1].

Implant slots were created in the bone model with the "boolean" operation in the Salome program. After the volumes and surfaces included in the boundary conditions were determined and named, all parts were combined.

Analysis was carried out in 28 configurations according to the vertical level difference of the left distal implant and its distance from the mesial implant. The vertical level of the left distal implant was examined at 1, 2 and 3 mm apically, 1, 2 and 3 mm occlusally, and at the same level as other implants. The distance of the left distal implant from the adjacent implant is modeled as 3.8, 4.7, 5.6 and 6.5 mm.

A convergence analysis was done on one of the cases with maximum displacement values of 0.01215 mm 0.01204 mm and 0.01207 mm for maximum element sizes of 0.3, 0.5 and 0.7 mm respectively. Also, maximum element size of 0.3 to 0.5 is standard in very similar studies [26–28].

## Boundary conditions and material properties

The material properties of cortical bone, trabecular bone, bars and implants were determined in the CalculiX software with reference to the literature (Table 1) [29,30]. All materials were considered linear, homogeneous and isotropic.

In the posterior part of the bone, the entire DOF (degree of freedom) was fixed by giving zero value (21). Assuming one hundred percent osseointegration at the bone and implant interface, the interface elements were combined with the "bounded contact" option [19,29–31]. A static load of 100 N was applied vertically to the left cantilever part of the bar [33,34].

## Results

The results obtained from the analysis were transferred to the Paraview (Kitware, Sandia National Laboratories, Los Alamos National Laboratory) program for interpretation.

Study models are divided into 4 main groups according to the distance of the left distal implant (*I1*) from the adjacent implant (*I2*). Each group is composed of subgroups whose vertical level of the *I1* varies from 3 mm apical to 3 mm coronal, in increments of one millimeter, compared to the other implants. These subgroups are named from A to G, from the most apical to the most coronal *I1* level. Minimum and maximum principal stress and strain values were measured for each model. The points where these values were highest in the cortical bone around the implant were evaluated. Since the stresses in trabecular bone were lower than cortical bone in all study models, they were not evaluated.

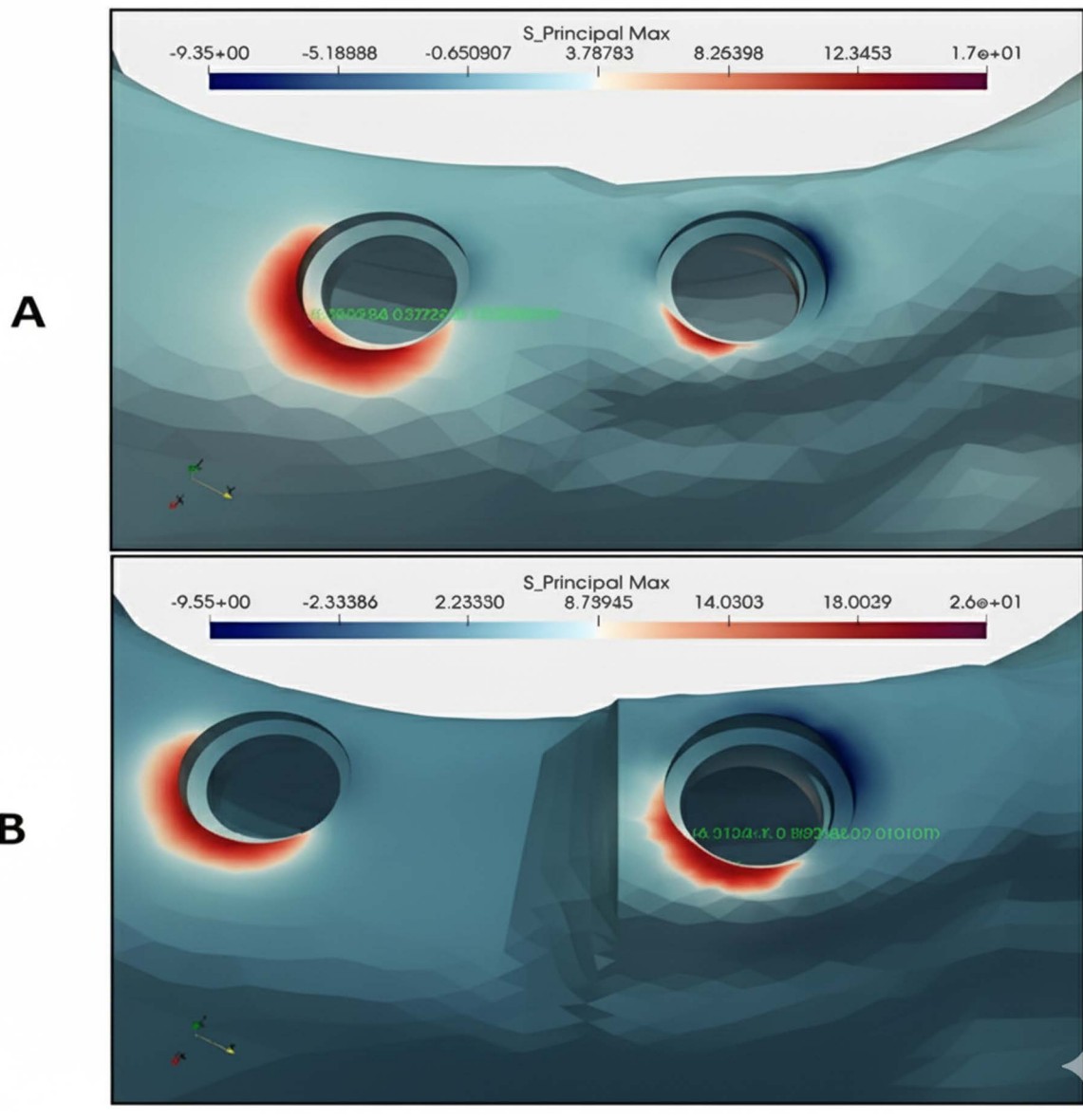

**Fig 2. A) Maximum principal stresses in group D. B) Maximum principal stresses in group G.**

**Table 1. Material properties.**

| | Young Modulus (Mpa) | Poisson`s Ratio |
|---|---|---|
| **Cortical [29]** | 13700 | 0.30 |
| **Trabecular [29]** | 1370 | 0.30 |
| **Titanium (implant, abutment) [30]** | 110000 | 0.33 |
| **Titanium grade 5 (bar) [30]** | 110000 | 0.28 |

### Maximum principal stress and strains in the cortical bone

The highest tensile stress and strains were found in the mesial part of *I2* or in some cases in the cortical bone around the *I2* (the maximum stress value was 25.7 MPa and the maximum strain value was 1578 microstrains (Fig 2). As *I1* was positioned more coronally, the tensile stress and strains in the mesial section increased. In addition, as the distance between *I1* and *I2* increased, the tensile stress and strains in the mesial section were observed to be higher. In all study models, similar maximum tensile stress and strain values were observed in the bone surrounding *I2* (average of 17 MPa for stress values and around 1000 microstrains for strain values). The values for each study group can be found in Fig 3.

### Minimum principal stress and strains in the cortical bone

The highest compressive stress and strains were found in the distal part of the cortical bone *I1* in all study models. In all main groups, the highest compression stress and strains were in subgroup D (where the *I1* is at the same level as the other implants (Fig 4)) (an average of −49 MPa for stress and around 3100 microstrains for strain values). It was observed that the stress and strain values decreased by up to twenty-two percent on average as *I1* was placed apically. It was also found that in models with *I1* placed coronally, minimum principal stress and strains were similar or lower than in models in which *I1* was at the same vertical level compared to other implants. In general, the Stress and strain values increased as the distance between *I1* and *I2* increased. The values for each study group can be found in Fig 5.

## Discussion

This study examines the strain and stress distribution in the bone resulting from the level and distance between the terminal implant and the adjacent implant in cases where bone is insufficient or when replacing a failed implant leads to suboptimal positioning. It represents the first investigation assessing the effects of vertical and horizontal distance variations on bone in four-implant-supported removable prostheses planned according to the RP-4 concept.

Terminal implants placed more coronally exhibited similar or lower compressive stress and strain in the bone compared to implants positioned at the same level. However, they led to a notable increase in tensile stress and strain in the surrounding bone. In contrast, models with more apically positioned terminal implants demonstrated a reduction in stress and strain. Additionally, increasing the distance between the terminal and adjacent implants resulted in higher stress and strain in the surrounding bone.

Considering the limitations of finite element analysis, this study suggests that in four-implant-supported, bar-retained removable prostheses, when the vertical positioning of the terminal implant is suboptimal due to bone irregularities or defects, tensile stress around the terminal implant may increase, but excessive stress and strain on the bone might still be avoided.

The assumption that bones are isotropic and homogeneous is generally made for the sake of theoretical simplicity and computational convenience. However, this assumption may not accurately reflect the true behavior of bone. In reality, bone is heterogeneous and anisotropic material, meaning it exhibits varying physical properties across different regions and demonstrates direction-dependent mechanical characteristics. Consequently, bone's mechanical response can differ depending on its orientation and location within the structure. In this study the bone was assumed to be isotropic and homogeneous which is a simplification of bone mechanics and studies have shown that this assumption can lead to underestimating the stress, specially under bending or mastication loads [35–37].

The finite element model developed in this study was based on geometrical, material, and loading parameters consistent with validated models reported in previous implant biomechanics research. The bone morphology, implant geometry, and boundary conditions were constructed according to established protocols in the literature [23–25,29–34]. Mesh refinement was performed iteratively, particularly in regions with potential stress concentration, to ensure convergence and numerical stability. The final mesh density (approximately 156,000 nodes and 32,000 tetrahedral elements) and the

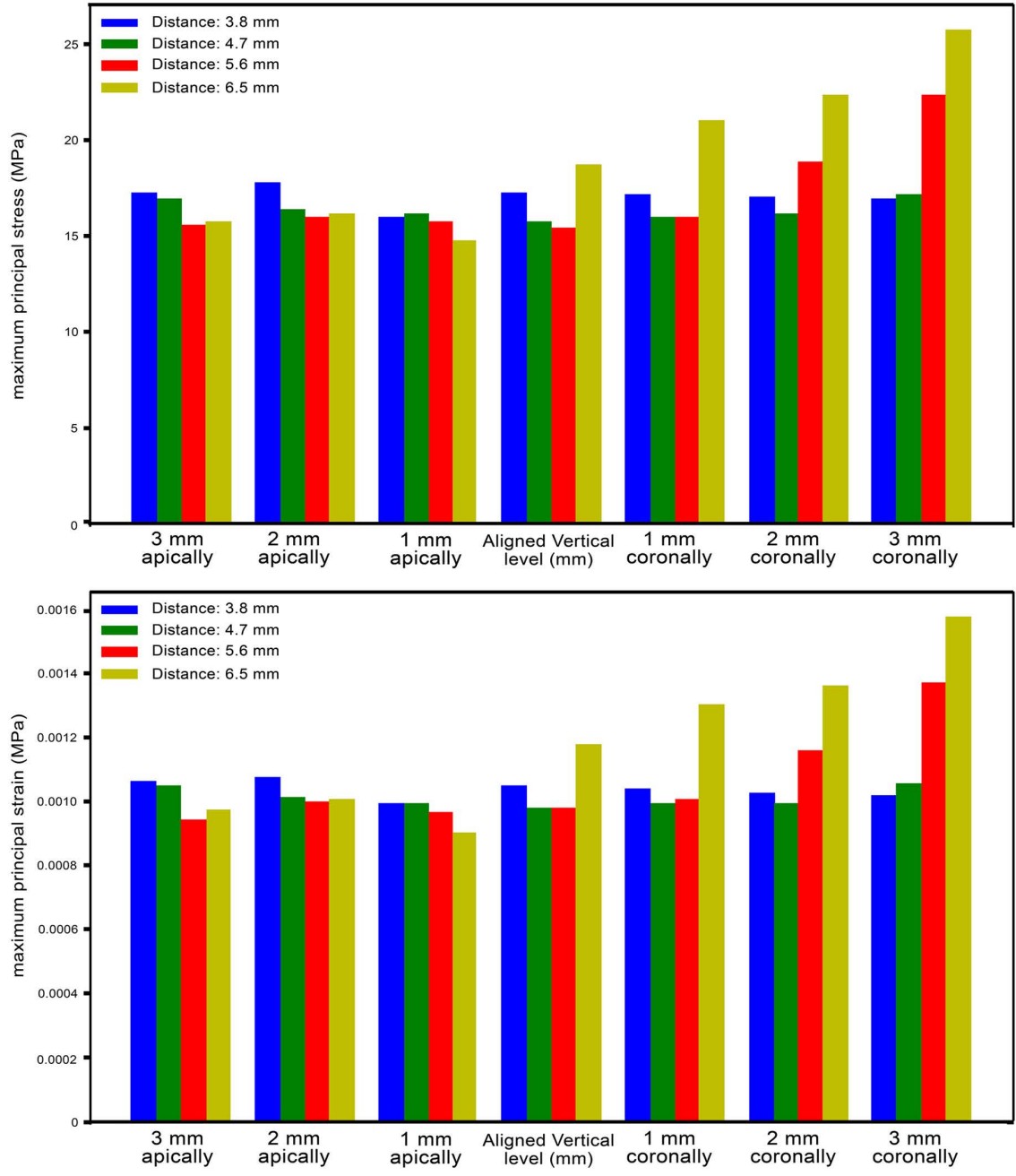

**Fig 3. Maximum principal strains for each study model and Maximum principal stresses for each study model.**

maximum element size on implant surfaces (0.3 mm) were in accordance with validated FE analyses in similar studies [16,29]. All materials were modeled as linear, homogeneous, and isotropic, following common assumptions in biomechanical simulations. Complete osseointegration between the implant and bone was assumed using bonded contact conditions, which is consistent with accepted modeling standards [19,29,31,32]. Although the model was not directly validated by experimental data, its configuration and parameter selection were based on previously verified approaches, ensuring that

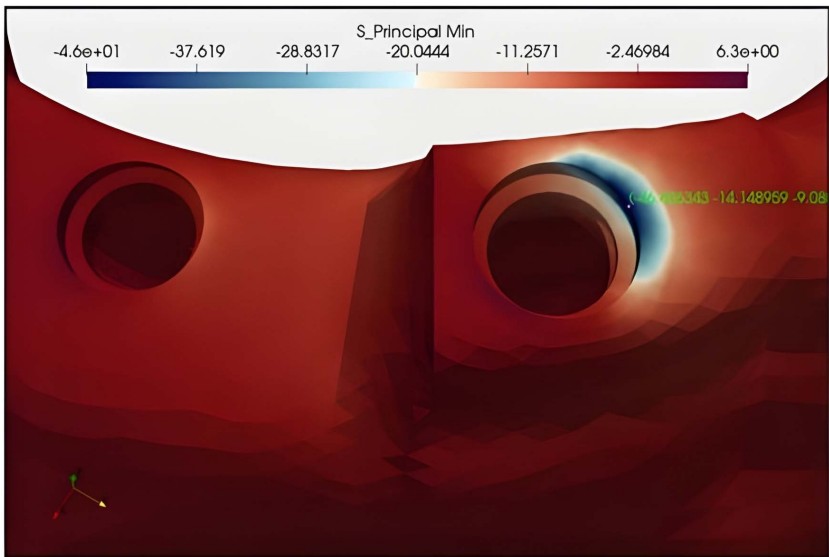

**Fig 4. Maximum principal stress in group G.**

the FE results are reliable for comparative and trend-based interpretations. Future research may incorporate experimental validation or patient-specific modeling to further confirm the quantitative accuracy of the present findings.

In a study investigating the effect of implant positioning on strain distribution around implants in four-implant-supported, telescopic-retained mandibular removable prostheses, implants were placed in three different acrylic jaw models: in the first molar and canine regions, the canine and second premolar regions, and the first premolar and lateral incisor regions. Strains were measured using the strain gauge method under a 50 N load applied to the prosthesis [38]. The highest strain values were observed in the model where implants were positioned in the second premolar and canine regions, while the lowest strains were recorded in models where implants were placed in the molar and canine regions [38]. Unlike current study, this study reported a decrease in stress as the distance between the terminal and adjacent implants increased. This discrepancy may be attributed to the fact that in these studies, the increased distance was achieved by positioning the distal implants further distally. In such cases, the anterior-posterior distance between implants increases, which can be expected to reduce stress levels. Tallarico et al. demonstrated that increasing the distance between implants can improve stress distribution [39]. However, in our study, an increase in distance was associated with higher stress and strain levels. This discrepancy may be due to differences in study designs or the increased cantilever effect.

In a study that examined the effects of vertical level differences of 1 mm and 3 mm between two implant-supported overdentures using three-dimensional finite element analysis, higher stress values were observed in the model with a 1 mm level difference. In contrast, the model with a 3 mm level difference exhibited lower stress values compared to the model with implants placed at the same level [19]. Another study evaluating stress distribution in two-implant-supported removable prostheses at different bone heights using three-dimensional finite element analysis reported similar stress values in models with and without level differences [20].

Our findings, when compared with previous studies, highlight the significant impact of level differences and implant positioning on stress distribution around implants. Notably, the increase in tensile stress in the bone surrounding more coronally positioned implants emerges as a factor that could influence implant success [19]. Furthermore, optimizing stress and strain levels through appropriate terminal implant positioning can be considered a critical factor for the long-term success of implant treatment. This underscores the importance of implant positioning as a key parameter in clinical practice.

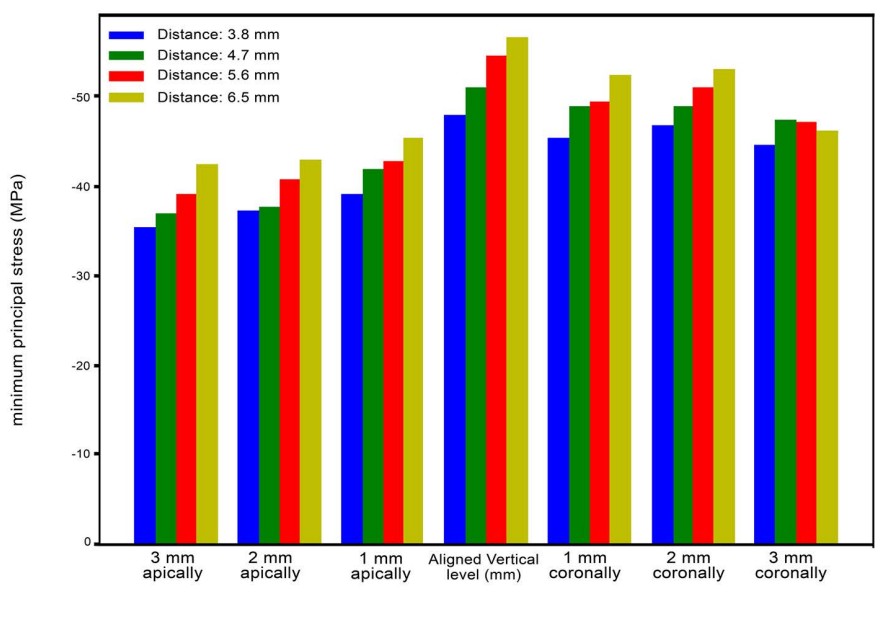

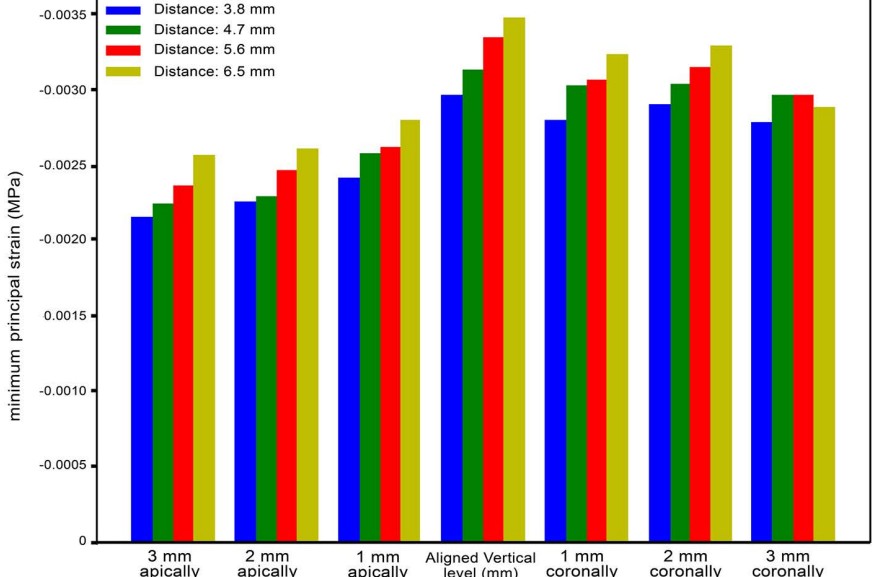

**Fig 5. Minimum principal stress for each study model and Maximum principal strains for each study model.**

Since RP-4 prostheses do not apply direct forces to the mucosa, they transmit forces directly to the bar component in specific clinical indications, such as cases with low chewing forces, absence of parafunctional habits, sufficient interarch space, and non-square-shaped jawbones. Due to these characteristics, RP-4 prostheses have been preferred in certain finite element analysis studies [40,41]. In our study, an RP-4 prosthesis was used, and to simplify calculations, a 100 N static vertical load was applied directly to the distal end of the left cantilever of the bar.

Bonnet et al. investigated the biomechanical effects of bone anisotropy in a four-implant-supported fixed bridge prosthesis in the mandible using three-dimensional finite element analysis [42]. In this study, muscle function was simulated,

and loads were applied to different regions of the prosthesis to mimic food material. The material properties of bone were analyzed in both isotropic and orthotropic models.

In the isotropic models, the lowest minimum principal stress and strain in the cortical bone around the terminal implant were calculated as −45.2 MPa and −3400 microstrains, respectively. In the orthotropic models, these values were −41.7 MPa and −2600 microstrains. The highest maximum principal stress and strain were observed around the anterior implant, with values of 4.9 MPa and 930 microstrains in the isotropic models and 5.9 MPa and 710 microstrains in the orthotropic models [42]. The ratio of the highest maximum principal stresses to the minimum principal stresses in Bonnet's study was lower than the corresponding ratio in the present study. This discrepancy may be attributed to the angled placement of the distal implants.

Frost suggested that excessive tensile stress in bone could lead to resorption and negatively impact implant success, advocating for implant positioning that minimizes tensile stresses [43]. The present study found that the highest tensile stress and strain occurred in the group where the left terminal implant was positioned most coronally and the adjacent implant was at the greatest distance, with values of 25.7 MPa and 0.0016, respectively. In groups where the distance between the left terminal and adjacent implant was minimal, the highest tensile stress and strain were located on the mesial aspect of the anterior implant, where the loads were applied, and the values were similar across models. As the distance between the left terminal implant and the adjacent implant increased and the terminal implant was placed more coronally, tensile stress and strain on the mesial aspect of the terminal implant also increased. This finding is consistent with recent finite element analyses reporting that inter-implant distance significantly alters stress concentration patterns and may shift mechanical load toward the terminal implant when spacing increases [44]. Additionally, clinical and radiologic evidence emphasizes that adequate horizontal spacing is essential for maintaining crestal bone, as insufficient spacing may predispose the inter-implant region to overload-related remodeling [45]. Notably, in the present study, when the distance between the left terminal implant and the adjacent implant reached 5.6 mm or more, and when the terminal implant was positioned at the same or a more coronal level, the tensile stress and strain around the terminal implant exceeded those of the adjacent implant. The highest compressive stress and strain were observed in the group where the terminal and adjacent implants were at the same level and had the greatest distance between them, with values of –56.7 MPa and –0.0035, respectively, while the lowest compressive values were consistently recorded around the terminal implant. In groups where the terminal implant was not positioned at the same level as the others, a reduction in the absolute compressive stress and strain values was observed. These findings align with evidence showing that vertical placement can modulate stress distribution, as subcrestal positioning has been associated with reduced peri-implant stress in comparative FEA models [26]. The von Mises stress distribution in the present study followed a pattern similar to compressive stress distribution.

Excessive strain within bone tissue may induce microcracks at the bone–implant interface. Pattin et al. reported critical damage thresholds for cortical bone of approximately 2500 microstrains in tension and 4000 microstrains in compression [46]. Consistent with these values, Sugiura et al. demonstrated that peri-implant strain levels approaching these limits may contribute to adverse biomechanical responses under varying bone densities and loading conditions, while Maghami et al. further confirmed that cortical bone exhibits microdamage initiation within similar strain ranges under both tensile and compressive loading [47,48]. In the present study, all cortical bone strain values remained below these thresholds, with maximum tensile and compressive strains of 1578 microstrains and 3473 microstrains, respectively. Nevertheless, masticatory forces differ substantially among individuals, and increased functional loading may elevate strain beyond these limits. Moreover, remaining below the reported thresholds does not definitively preclude the possibility of bone remodeling or resorption around the implant. Consequently;

• In bar-retained overdenture treatments supported by four implants, when force was applied to the distal part of the cantilever, positioning the terminal implant at a higher vertical level did not increase compressive strain and stress in the surrounding bone. However, a significant increase in tensile strain and stress around the terminal implant was observed.

- Positioning the terminal implant more apically compared to the other implants reduced strain and stress in the surrounding bone. In general, as the distance between the terminal implant and the adjacent implant increased, strain and stress in the surrounding bone also increased.

- In four-implant-supported bar-retained overdenture treatments, the highest compressive stress and strain under distal cantilever loading were observed in the distal aspect of the bone surrounding the terminal implant on the same side.

- The highest tensile stress and strain were observed in the mesial aspect of the bone surrounding both the terminal and adjacent implants.

- Considering the limitations of finite element analysis studies, the findings suggest that in four-implant-supported bar-retained overdenture treatments, when the vertical level of the terminal implant is not ideal due to bone irregularities or defects, excessive stress and strain in the surrounding bone may not necessarily occur.

## Conclusion

Within the limitations inherent to finite element analysis, the present study suggests that in four-implant-supported, bar-retained overdenture treatments, suboptimal vertical positioning of the distal implant—arising from bone irregularities or defects—may lead to localized increases in stress and strain within the surrounding bone; however, these increases appear to remain within physiological limits. From a clinical perspective, this finding implies that minor deviations in the vertical level of the terminal implant are unlikely to jeopardize peri-implant bone integrity or the long-term biomechanical stability of the overdenture. In contrast, greater interimplant distances between the distal and adjacent implants were associated with higher stress concentrations, underscoring the importance of optimal implant spacing to minimize the risk of bone overload and mechanical complications. Consequently, meticulous preoperative planning and precise implant positioning remain essential to achieve favorable stress distribution and long-term success in four-implant-supported bar-retained overdenture treatments.

## Supporting information

**S1 Data. Finite element model of the mandibular structure.**
(DOCX)

## Author contributions

**Conceptualization:** Danesh Ghahramanimarangalou, Cennet Neslihan Eroglu.

**Data curation:** Danesh Ghahramanimarangalou.

**Funding acquisition:** Danesh Ghahramanimarangalou.

**Investigation:** Danesh Ghahramanimarangalou.

**Methodology:** Danesh Ghahramanimarangalou, Cennet Neslihan Eroglu.

**Project administration:** Cennet Neslihan Eroglu.

**Software:** Danesh Ghahramanimarangalou.

**Supervision:** Nurullah Turker, Cennet Neslihan Eroglu.

**Validation:** Nurullah Turker.

**Visualization:** Cennet Neslihan Eroglu.

**Writing – original draft:** Danesh Ghahramanimarangalou, Cennet Neslihan Eroglu.

**Writing – review & editing:** Danesh Ghahramanimarangalou, Nurullah Turker, Cennet Neslihan Eroglu.

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
