## [Decision Letter · Decision Letter 0]

14 Oct 2025

Dear Dr. Eroglu,

We look forward to receiving your revised manuscript.

Kind regards,

Mohmed Isaqali Karobari, BDS, MScD.Endo, Ph.D. Endo, FDS, FPFA, FICD, MFDS

Academic Editor

PLOS ONE

Journal Requirements:

3. We note that your Data Availability Statement is currently as follows: “All relevant data are within the manuscript and its Supporting Information files.”

Additional Editor Comments:

Dear Authors,

Please carefully read all the comments provided by the reviewers and address them accordingly, making the necessary changes in the revised manuscript.

Best regards and keep well

Reviewer's Responses to Questions

**Comments to the Author**

1. Is the manuscript technically sound, and do the data support the conclusions?

Reviewer #1: Yes

Reviewer #2: Partly

2. Has the statistical analysis been performed appropriately and rigorously?

Reviewer #1: N/A

Reviewer #2: N/A

3. Have the authors made all data underlying the findings in their manuscript fully available?

Reviewer #1: Yes

Reviewer #2: No

4. Is the manuscript presented in an intelligible fashion and written in standard English?

Reviewer #1: Yes

Reviewer #2: Yes

Reviewer #1: Dear Authors,

The study was planned well and driven flawlessly. However, some concerns need to be done before publication. Kindly discuss contemporary literature in the discussion section instead of outdated researches in this particular field. Please express possible clinical outcomes in light of your study. My decision is that the manuscript will be accepted for publication after minor revision.

Best regards.

Reviewer #2: This study used 3D finite element analysis to evaluate how the horizontal distance and vertical misalignment between distal and adjacent implants in four-implant bar-retained overdentures affect bone stress and strain. Results showed that vertical discrepancies had minimal biomechanical impact, whereas greater inter-implant distances increased stress in surrounding bone. This study holds scientific merit, and the authors deserve recognition for their diligent work. However, several key issues need to be addressed before the manuscript can be considered for publication.

Major comments:

1. Regarding the finite element simulation, the manuscript needs a clear statement defining the accuracy of the FE model. The reviewer thought that the FE model used in this study needs to be validated before use. Alternatively, please consider adding one paragraph to the “Discussion” section that describes the reliability of the FE model used in this study.

2. The manuscript mentions that tetrahedral meshes were generated using the NETGEN algorithm and that repeated mesh refinement was performed in regions with potential singularities. However, no information is provided on whether a formal mesh convergence analysis was conducted to verify the stability and accuracy of the finite element results.

3. Could the authors clarify if a mesh convergence test was performed, and if not, please discuss how the chosen mesh density ensures numerical reliability of the results?

4. The authors are encouraged to include a complete illustration of the finite element model used in this study. Please provide a figure showing the overall geometry of the mandibular model, the implant-bar assembly, and the boundary conditions. In addition, it would be highly beneficial to display the different components and material regions (e.g., cortical bone, trabecular bone, implants, and bar structure) within the model to help readers better understand how the simulation was constructed and how the interfaces were defined.

5. In the finite element model, the cortical and trabecular bones were assumed to be linear, homogeneous, and isotropic materials. However, in reality, bone exhibits anisotropic and heterogeneous mechanical behavior. This simplification may significantly influence the stress and strain distribution results. The authors are advised to discuss how this assumption might affect the accuracy of their simulation outcomes and whether it could alter the biomechanical interpretation of the findings.

Minor comments:

1. The abstract lacks supporting evidence. The authors must provide sufficient quantitative data to support their claims.

2. The resolution of Figures 1–5 is too low, making it difficult to clearly observe the details presented in each image. The authors are requested to provide higher-resolution and clearer versions of all figures to improve the readability and visual quality of the manuscript.

3. The discussion section mentions the limitations of the study only briefly. It is recommended that the authors expand on the study’s limitations and present them as a separate paragraph at the end of the Discussion section. Specifically, the authors may elaborate on aspects such as the idealized geometry of the model, the assumption of isotropic bone properties, the absence of in vitro or clinical validation, and the lack of a mesh convergence test. A more detailed and structured limitations paragraph would enhance the scientific transparency and rigor of the paper.

**Do you want your identity to be public for this peer review?** For information about this choice, including consent withdrawal, please see our Privacy Policy

Reviewer #1: No

Reviewer #2: No

---

## [Author Response · Author response to Decision Letter 1]

24 Nov 2025

The requested revisions to the manuscript (PONE-D-25-47053 Impact of Distance and Vertical Placement of Distal Implants on Bone Mechanics in Bar-retained Overdenture Treatment: A 3d Finite Element Study) have been completed, and all issues raised by the reviewers have been addressed. The corrections specified for “journal requirements” have also been made. We sincerely thank you for taking the time to evaluate our study.

Reviewer #1: Dear Authors,

The study was planned well and driven flawlessly. However, some concerns need to be done before publication. Kindly discuss contemporary literature in the discussion section instead of outdated researches in this particular field. Please express possible clinical outcomes in light of your study. My decision is that the manuscript will be accepted for publication after minor revision.

Thank you very much for taking the time to review our manuscript. Recent studies have been incorporated, and the conclusion section has been revised to better reflect the potential clinical implications of the findings. All modifications made in accordance with the reviewer’s suggestions have been highlighted in green within the manuscript.

Reviewer #2: This study used 3D finite element analysis to evaluate how the horizontal distance and vertical misalignment between distal and adjacent implants in four-implant bar-retained overdentures affect bone stress and strain. Results showed that vertical discrepancies had minimal biomechanical impact, whereas greater inter-implant distances increased stress in surrounding bone. This study holds scientific merit, and the authors deserve recognition for their diligent work. However, several key issues need to be addressed before the manuscript can be considered for publication.

Thank you very much for taking the time to review our manuscript.

Major comments:

1. Regarding the finite element simulation, the manuscript needs a clear statement defining the accuracy of the FE model. The reviewer thought that the FE model used in this study needs to be validated before use. Alternatively, please consider adding one paragraph to the “Discussion” section that describes the reliability of the FE model used in this study.

The fifth paragraph of the Discussion section (highlighted in yellow) has been added to address this revision.

2. The manuscript mentions that tetrahedral meshes were generated using the NETGEN algorithm and that repeated mesh refinement was performed in regions with potential singularities. However, no information is provided on whether a formal mesh convergence analysis was conducted to verify the stability and accuracy of the finite element results.

A convergence analysis was done on one of the cases with maximum displacement values of 0.01215 mm 0.01204 mm and 0.01207 mm for maximum element sizes of 0.3, 0.5 and 0.7 mm respectively. Also maximum element size of 0.3 to 0.5 is standard in very similar studies (1-3).

References

1. Di Pietro N, Capparé P, Nagni M, et al. Finite element analysis (FEA) of the stress and strain distribution in Cone-Morse implant–abutment connection implants placed equicrestally and subcrestally. Appl Sci (Basel). 2023;13(14):8147. doi:10.3390/app13148147

2. Ceddia A, Assenza B, Bollero P, et al. Prediction of dental implant primary stability with cone beam computed tomography and finite element analysis. Materials (Basel). 2025;18(7):1625. doi:10.3390/ma18071625

3. Vautrin A, Bouchard P, Rieger D, et al. Homogenized finite element simulations can predict the mechanical response of bone-implant systems. J Mech Behav Biomed Mater. 2024;159:106158. doi:10.1016/j.jmbbm.2024.106158

3. Could the authors clarify if a mesh convergence test was performed, and if not, please discuss how the chosen mesh density ensures numerical reliability of the results?

A convergence analysis was done.

4. The authors are encouraged to include a complete illustration of the finite element model used in this study. Please provide a figure showing the overall geometry of the mandibular model, the implant-bar assembly, and the boundary conditions. In addition, it would be highly beneficial to display the different components and material regions (e.g., cortical bone, trabecular bone, implants, and bar structure) within the model to help readers better understand how the simulation was constructed and how the interfaces were defined.

Fig 1 (new added) shows the overall geometry of the mandibular model, the implant-bar assembly, and the boundary conditions applied during the simulation.

5. In the finite element model, the cortical and trabecular bones were assumed to be linear, homogeneous, and isotropic materials. However, in reality, bone exhibits anisotropic and heterogeneous mechanical behavior. This simplification may significantly influence the stress and strain distribution results. The authors are advised to discuss how this assumption might affect the accuracy of their simulation outcomes and whether it could alter the biomechanical interpretation of the findings.

In response to your valuable comment, this topic has been addressed in the fourth paragraph, highlighted in yellow, of the Discussion section.

Minor comments:

1. The abstract lacks supporting evidence. The authors must provide sufficient quantitative data to support their claims.

The abstract has been revised and the changes are highlighted in yellow.

2. The resolution of Figures 1–5 is too low, making it difficult to clearly observe the details presented in each image. The authors are requested to provide higher-resolution and clearer versions of all figures to improve the readability and visual quality of the manuscript.

We have further improved the quality of the images, and we hope they now meet the expected standards.

3. The discussion section mentions the limitations of the study only briefly. It is recommended that the authors expand on the study’s limitations and present them as a separate paragraph at the end of the Discussion section. Specifically, the authors may elaborate on aspects such as the idealized geometry of the model, the assumption of isotropic bone properties, the absence of in vitro or clinical validation, and the lack of a mesh convergence test. A more detailed and structured limitations paragraph would enhance the scientific transparency and rigor of the paper.

The Discussion section has been expanded to provide more comprehensive coverage of the topic. We believe that the revisions made in response to the major comments have inevitably led to an expansion of the Limitations section as well.

---

## [Decision Letter · Decision Letter 1]

11 Dec 2025

Impact of Distance and Vertical Placement of Distal Implants on Bone Mechanics in Bar-retained Overdenture Treatment: A 3d Finite Element Study

PONE-D-25-47053R1

Dear Dr. Eroglu,

We’re pleased to inform you that your manuscript has been judged scientifically suitable for publication and will be formally accepted for publication once it meets all outstanding technical requirements.

Kind regards,

Mohmed Isaqali Karobari, BDS, MScD.Endo, Ph.D. Endo, FDS, FPFA, FICD, MFDS

Academic Editor

PLOS One

Additional Editor Comments (optional):

Dear Authors,

The authors have addressed all the comments and suggestions provided by the reviewers, and the manuscript has undergone significant improvement. I would like to congratulate the authors and wish them all the very best in their future endeavours.

Best regards and keep well.

Reviewers' comments:

Reviewer's Responses to Questions

**Comments to the Author**

Reviewer #1: All comments have been addressed

Reviewer #2: All comments have been addressed

2. Is the manuscript technically sound, and do the data support the conclusions?

Reviewer #1: Yes

Reviewer #2: Yes

3. Has the statistical analysis been performed appropriately and rigorously?

Reviewer #1: N/A

Reviewer #2: N/A

4. Have the authors made all data underlying the findings in their manuscript fully available?

Reviewer #1: Yes

Reviewer #2: Yes

5. Is the manuscript presented in an intelligible fashion and written in standard English?

Reviewer #1: Yes

Reviewer #2: Yes

Reviewer #1: Dear Authors,

Thank you to be addressed all comments. The manuscript can be accepted for publication.

Best regards.

Reviewer #2: (No Response)

**Do you want your identity to be public for this peer review?** For information about this choice, including consent withdrawal, please see our Privacy Policy

Reviewer #1: No

Reviewer #2: No

---

## [Editor Report · Acceptance letter]

PONE-D-25-47053R1

PLOS One

Dear Dr. Eroglu,

I'm pleased to inform you that your manuscript has been deemed suitable for publication in PLOS One. Congratulations! Your manuscript is now being handed over to our production team.

Kind regards,

on behalf of

Prof Dr. Mohmed Isaqali Karobari

Academic Editor

PLOS One